# Variations in the Composition of Human Milk Oligosaccharides Correlates with Effects on Both the Intestinal Epithelial Barrier and Host Inflammation: A Pilot Study

**DOI:** 10.3390/nu14051014

**Published:** 2022-02-28

**Authors:** Richard Y. Wu, Steven R. Botts, Kathene C. Johnson-Henry, Eva Landberg, Thomas R. Abrahamsson, Philip M. Sherman

**Affiliations:** 1Cell Biology Program, Research Institute Hospital for Sick Children, Toronto, ON M5G 0A4, Canada; you.wu@mail.utoronto.ca (R.Y.W.); steven.botts@mail.utoronto.ca (S.R.B.); kathene.johnson-henry@sickkids.ca (K.C.J.-H.); 2Division of Gastroenterology, Hepatology and Nutrition, Department of Paediatrics, University of Toronto, Toronto, ON M5S 1A1, Canada; 3Department of Clinical Chemistry, Linköping University, 58183 Linköping, Östergötland, Sweden; eva.landberg@regionostergotland.se; 4Department of Biomedical and Clinical Sciences, Linköping University, 58183 Linköping, Östergötland, Sweden; thomas.abrahamsson@liu.se; 5Department of Pediatrics, Linköping University, 58183 Linköping, Östergötland, Sweden

**Keywords:** inflammation, human milk oligosaccharides, intestinal epithelial barrier

## Abstract

Background: Human milk oligosaccharides are complex, non-digestible carbohydrates that directly interact with intestinal epithelial cells to alter barrier function and host inflammation. Oligosaccharide composition varies widely between individual mothers, but it is unclear if this inter-individual variation has any impact on intestinal epithelial barrier function and gut inflammation. Methods: Human milk oligosaccharides were extracted from the mature human milk of four individual donors. Using an in vitro model of intestinal injury, the effects of the oligosaccharides on the intestinal epithelial barrier and select innate and adaptive immune functions were assessed. Results: Individual oligosaccharide compositions shared comparable effects on increasing transepithelial electrical resistance and reducing the macromolecular permeability of polarized (Caco-2Bbe1) monolayers but exerted distinct effects on the localization of the intercellular tight junction protein zona occludins-1 in response to injury induced by a human enteric bacterial pathogen *Escherichia coli*, serotype O157:H7. Immunoblots showed the differential effects of oligosaccharide compositions in reducing host chemokine interleukin 8 expression and inhibiting of p38 MAP kinase activation. Conclusions: These results provide evidence of both shared and distinct effects on the host intestinal epithelial function that are attributable to inter-individual differences in the composition of human milk oligosaccharides.

## 1. Introduction

Human milk is the gold standard for infant nutrition up to six months of age [1,2]. Human milk not only provides essential micro- and macronutrients, but also contains a plethora of bioactive compounds [1]. Following carbohydrate lactose and lipids, human milk oligosaccharides (HMOs) constitute the third most abundant component of mature human milk in concentrations between 5 and 15 g/L [3]. After consumption by the infant, ingested HMOs traverse the gastrointestinal tract mostly intact to the large intestine, where they exert prebiotic effects via promoting the growth of beneficial microbes that are present in the resident colonic microbiota [4]. HMOs also serve as receptor analogues to the block binding of enteric pathogens and their toxins to mucosal surfaces [5,6] and can directly activate cell surface receptors to influence various functions of the intestinal epithelial barrier [7].

Unlike plant-derived glycans or animal sources of milk oligosaccharides, HMOs are highly complex oligosaccharides notable for marked variability, with up to 150 structures that differ between lactating individuals [8]. Structurally, HMOs share a common lactose core, which can be elongated from five individual monosaccharides: glucose, galactose, N-acetylglucosamine, fucose and sialic acid [9]. Much of the compositional variation is influenced by the maternal expression of Secretor and Lewis genes [10], geography [11] and the maternal diet and BMI during lactation [12].

Structural variation in HMOs may have clinical implications in meeting the health needs of infants fed human milk. For instance, fucosylated HMOs inhibit the binding of rotavirus to receptors present on mucosal surfaces [13], sialyated HMOs correlate with infant growth [14] and HMOs such as disialyllacto-N-tetraose are associated with the prevention of necrotizing enterocolitis in preterm infants [15,16]. Despite such associations, identifying structure–function relationships for HMOs have been challenging. To date, most studies have evaluated the effects of a single HMO without considering the functional relevance of inter-individual variations in the composition of HMOs. Using RNA sequencing, we recently demonstrated that five synthetic HMOs each induced a unique set of host cell transcriptomic changes with little overlap between individual structures [17]. In the present study, we performed in vitro functional comparisons of HMOs extracted from the mature human milk of four individual mothers. Human intestinal epithelial (Caco-2Bbe1) cells were exposed to each of the milk preparations before markers of epithelial barrier integrity, inflammation and host cell signaling were measured. The findings demonstrate that distinct compositions of HMOs elicit both common and differing host epithelial cell responses.

## 2. Materials and Methods

### 2.1. Human Milk Collection

Donated surplus human milk samples from four individual unidentified donors (approx. 25–40 bottles of 35–50 mL per mother) were collected from the Department of Nutritional Services at the Hospital for Sick Children (Toronto, ON, Canada) and stored at −80 °C. To eliminate batch-to-batch variations during collection over time, samples were immediately thawed (4 °C), pooled for each individual and re-aliquoted into 50 mL conical tubes. Prior to HMO isolation, frozen samples were pasteurized using the Holder protocol where individual aliquots were submersed in a 62.5 °C water bath for 30 min, as previously described [18].

### 2.2. Isolation of HMOs

HMOs from individual mothers were isolated, as previously described [15]. Following pasteurization, human milk samples were centrifuged (4000× *g* at 4 °C for 45 min) to remove the topmost lipid content. Remaining aqueous samples were treated with ice-cold acetone (Sigma-Aldrich, St. Louis, MO, USA) overnight to precipitate milk proteins. Acetone content was then evaporated using roto-evaporation. Carbohydrate-rich HMO preparations were then filtered using Bio-Rad P2 polyacrylamide gel (Bio-Rad, Hercules, CA, USA) packed into 0.7 × 20 cm glass columns (VWR, Mississauga, ON, Canada) to remove lactose and salts. HMOs from each individual mother were collected, roto-evaporated and weighed before solubilizing in sterile phosphate-buffered saline (PBS; Gibco, Waltham, MA, USA) to a concentration of 200 mg/mL. Each individual HMO preparation was stored at −20 °C until the day of experimentation.

### 2.3. Composition Analysis of HMOs

Relative concentrations of the 16 major HMOs were analyzed based on an established protocol using high-performance anion-exchange chromatography coupled with pulsed amperometric detection (HPAEC–PAD) (ICS-3000, Dionex, Sunnyvale, CA, USA) [19]. Oligosaccharide separation was performed under a 0.5 mL/min flow rate at 20, 30 or 40 °C. Elution gradient programs were selected based on oligosaccharide acidity. Neutral oligosaccharides were eluted with sodium hydroxide (20 mM; 0–5 min), followed by increasing concentrations of sodium acetate (0–25 mM for 25 min). Acidic oligosaccharides were eluted with sodium hydroxide (100 mM) followed by increasing concentrations of sodium acetate (20–80 mM for 5–30 min; 80–200 mM for 30–40 min). HMOs were quantified by comparing retention times and peak area with HMO standards of known concentrations (Dextra Laboratories, Reading, UK). The hierarchal clustering of individual mother HMO preparations was performed with STAMP using the unweighted pair group method with the arithmetic mean [20].

### 2.4. Bacterial Culture

Enterohemorrhagic *Escherichia coli* (EHEC) serotype O157:H7, strain CL-56 [21] bacterial cultures were stored at −80 °C, fast-thawed and then plated on Columbia 5% blood agar plates (BBL, Toronto, ON, Canada) overnight at 37 °C. EHEC colonies on blood agar plates were stored at 4 °C. On the day of the experiment, Caco-2Bbe1 cells were pretreated with individual pooled HMOs at 20 mg/mL for 16 h overnight at 37 °C. In parallel, single colonies were cultured in Penassay broth (Gibco) and grown overnight at 37 °C for 16 h, and then sub-cultured into fresh Penassay broth (Gibco) for 3 h. Following overnight HMO pre-treatment, subcultured EHEC O157:H7 was centrifuged (500 rpm, 5 min) and then added into the apical compartment of polarized epithelial cell monolayers at a multiplicity of infection (MOI) of 100 to 1 for a total of 4 h for Transwell TER and dextran flux assays. For MAPK and NFκβ signaling assays, EHEC O157:H7 was added onto confluent Caco-2Bb1 monolayers grown in 12-well plates (Corning, Mississauga, ON, Canada) at an MOI of 10:1 for a total of 3 h of co-incubation. For the mRNA and protein expression experiment, EHEC was added onto Caco-2Bbe1 monolayers grown in 24-well plates at MOI of 100:1 for a total of 4 h.

### 2.5. Cell Culture and Measures of Epithelial Barrier Integrity

Caco-2Bbe1 cells, a differentiated subclone of Caco-2 cells, were purchased from the American Type Culture Collection (ATCC, Mannasas, VA, USA) and grown using previously described methods [21]. For culturing in Transwells, Caco-2Bbe1 cells were harvested with trypsin-EDTA and seeded onto filter supports (6.5 mm diameter, Corning) at a confluence of 10^5^ cells per insert. Cells were grown until confluence, as indicated by a TER of ≥800 Ω/cm^2^. Polarized Caco-2Bbe1 monolayers were submersed in 200 μL medium in the apical compartment, and 1000 μL medium in the basolateral side, which were changed every 2 days. Cells were treated either with medium alone or challenged with EHEC O157:H7 in the absence or presence of individual HMO preparations.

### 2.6. Transepithelial Electrical Resistance (TER)

TER was measured at baseline once Caco-2Bbe1 monolayers had reached a TER ≥ 800 Ω/cm^2^. During pre-treatment with HMOs, the apical culture medium was aspirated and replaced with 20 mg/mL of HMOs suspended in tissue culture medium. HMOs were dissolved in pre-warmed tissue culture medium and incubated overnight (16 h, 37 °C). EHEC O157:H7 was added into the apical compartment of Transwells at an MOI of 100:1 for 4 h, 37 °C following pre-treatment with HMOs (20 mg/mL, 16 h). TER measurements were taken after 4 h of bacterial incubation by using chopstick electrodes (Millicell, Millipore, Etobicoke, ON, Canada). TER after bacterial challenge in the absence and presence of HMOs was calculated as a percentage of the baseline TER relative to control, uninfected cell monolayers. All experiments were performed using technical duplicates.

### 2.7. FITC-Dextran Permeability Assay

The assessment of monolayer permeability to an intact macromolecule was performed by employing the fluorescein-labeled dextran assay using previously described methods [21]. Briefly, following EHEC O157 challenge and final TER recordings, apical compartments were washed three times with warm PBS and then replaced with cultured media containing 10-kDa fluorescein-labeled dextran at a concentration of 100 μg/mL. Transwells were then incubated for 5 h at 37 °C and 50 μL aliquots from basolateral compartments then collected, and the signal intensities were measured using an infrared imaging system (Odyssey^®^, LI-COR Biosciences, Lincoln, NE, USA) at a wavelength of 700 nm. Quantified signal intensities were calculated as absolute dextran quantities, in nanograms, using a serial dilution curve.

### 2.8. Immunofluorescence Microscopy

Caco-2Bbe1 monolayers were grown in 12-well plates on glass coverslips, and immunofluorescence was performed using a previous protocol [21]. In brief, cell monolayers were fixed with 10% paraformaldehyde at 4 °C overnight, permeabilized using 0.1% Triton X-100, and non-specific binding was then blocked with 3% bovine serum albumin (Sigma, St. Louis, MO, USA). Cells were then washed with cold PBS and incubated with rabbit anti-ZO-1 antibody (Life Technologies, Invitrogen, Mississauga, ON, Canada) and goat Alexa-fluor 488-conjugated anti-rabbit IgG antibodies (Life Technologies). Cells were then mounted onto glass slides and visualized using a Leica DMI6000B fluorescence microscope and a DFC 360FX camera lens (Leica Microsystems, Concord, ON, Canada). Images were taken at random five times per slide at 20x of original magnification.

### 2.9. qPCR for mRNA Expression

Following challenge with EHEC O157:H7, cells were washed three times with PBS and RNA extracted using TriZOL (Thermo Fisher, Mississauga, ON, Canada) as per the manufacturer’s protocol. cDNA was generated using an iScript cDNA synthesis kit (Bio-Rad, Mississauga, ON, Canada) based on the manufacturer’s protocol. qPCR was performed as previously described using a CFX96 C1000 Thermal Cycler (Bio-Rad) using iQ SYBR Green Supermix co-mixed with 500 ng template cDNA [19]. Expression levels were calculated using the ∆∆C_t_ method and normalized to *GAPDH* and *β-actin* housekeeping genes. All experiments were performed using technical duplicates. The primer sequences used were:*GAPDH*, ACCCACTCCTCCACCTTTGAC (forward), CCACCACCCTGTTGCTGTAG (reverse);*Β-actin*, CTGGAACGGTGAAGGTGACA (forward), AAGGGACTTCCTGTAACAATGCA (reverse).*ZO-1*, GAATGATGGTTGGTATGGTGCG (forward), TCAGAAGTGTGTCTACTGTCCG (reverse);*Claudin-1*, AGCTGGCTGAGACACTGAAGA (forward), GAGAGGAAGGCACTGAACCA (reverse);*Muc1*, CCTCACAGTGCTTACAGTTGTT (forward), AGTAGTCGGTGCTGGGATCT (reverse);*Muc2*, TGTAGGCATCGCTCTTCTCA (forward), GACACCATCTACCTCACCCG (reverse);*Tgfβ*, CGGAGTTGTGCGGCAGTGGT (forward), GGCCGGTAGTGAACCCGTTGATG (reverse);*Il-10*, AGGAGGTGATGCCCCAAGCTGA (forward), ATCGATGACAGCGCCGTAGCCTC (reverse);*CXCL-8*, ACTGAGAGTGATTGAGAGTGGAC (forward), AACCCTCTGCACCCAGTTTTC (reverse);*IL-18*, TGCCCTCCTGGCTGCCAACT (forward), TCAGCAGCCATCTTTATTCCTGCG (reverse).

### 2.10. Immunoblotting

Anti-ZO-1, anti-claudin-1 and anti-β-actin primary antibodies were purchased from Invitrogen (Burlington, ON, Canada); anti-GAPDH was purchased Santa-Cruz (Mississauga, ON, Canada); anti-phospho-ERK 1/2, anti-ERK 1/2, anti-phospho-P38, anti-P38 and anti-IκBα primary antibodies were purchased from New England Biolabs (NEB, Whitby, ON, Canada). Immunoblotting techniques were performed using previously described methods [21].

### 2.11. Statistical Analyses

Results were expressed as means ± SEM for continuous variables. Statistical testing was performed using GraphPad Prism 6.0 (Prism). Pairwise comparisons were made using the Student’s t-test, and multiple group comparisons were performed using ANOVA with Bonferroni post hoc correction. The statistical significance level was determined as a *p* value of less than 0.05.

## 3. Results

### 3.1. HMO Composition in Individual Mothers

To assess variation in the HMO composition, donated mature human milk samples were collected from four lactating healthy mothers at the Hospital for Sick Children in Toronto, Canada. Individual samples from each mother were collected over the first month post-partum, pooled and then extracted for HMOs by using previously described methods [15]. Oligosaccharide composition was measured using high-performance anion-exchange chromatography with pulsed amperometric detection (HPAEC–PAD) [19]. As shown in Figure 1A, all four donors were Secretors, but there was variation in the HMO composition across each of the four mothers. Mothers 2 and 3 had higher abundances of the HMOs with α1-2-fucosylation (2′-FL, LDFT and LNFPI) (Figure 1A,B). In contrast, mothers 1 and 4 had lower levels of α1-2-fucosylated HMOs and a higher abundance of neutral HMOs (LNT) (Figure 1C). Mother 3 was Lewis-negative (*Le*-) due to the absence of α1-4-fucosylated HMOs [22]. The hierarchical clustering of HMO composition identified two major clusters (Figure 1D): a first cluster involving mothers 1 and 4 and a second cluster with mothers 2 and 3. Taken together, these results demonstrate that individuals expressed distinct HMO blends.

### 3.2. HMOs Promote Intestinal Epithelial Barrier Integrity in an Individual-Dependent Manner

To compare the functional effects of individual HMO blends, an epithelial injury model was employed, in which Caco-2 cells were pre-treated with individual HMO blends (16 h; 20 mg/mL) and then challenged with enterohemorrhagic *Escherichia coli* (EHEC), serotype O157:H7 strain CL-56 (MOI 100:1; 4 h) [21,23]. As shown in Figure 2A, EHEC challenge completely disrupted ZO-1 distribution, whereby the normal continuous, circumferential localization was changed into distinct puncta (Figure 2A). HMOs from individuals 1 and 4, but not individuals 2 or 3, completely prevented EHEC-induced ZO-1 redistribution (Figure 2A).

To validate these structural changes in intercellular tight junction morphology, the effects of HMOs on intestinal epithelial barrier function were tested by measuring both transepithelial electrical resistance (TER) and transcellular permeability using a 10 kDa fluorescein isothiocyanate-dextran probe [21]. In both the absence and presence of EHEC O157:H7, each of the HMO blends significantly increased Caco-2Bbe1 TER compared to the baseline levels, with HMOs prepared from individual #4 having the greatest effect (Figure 2B). As shown in Figure 2C, the dextran flux assay showed that EHEC-induced increases in macromolecule permeability were reduced in polarized epithelia pretreated with HMOs. Taken together, these results indicate that individual HMO blends share barrier-protective properties, regardless of composition, but vary in their ability to protect against bacterial pathogen-induced disruption in the localization of intercellular tight junction proteins.

### 3.3. Individual HMO Blends Alter Barrier-Related Gene Expression

To determine how changes in epithelial barrier function were related to exposure to various HMO preparations, we next measured mRNA levels of *Muc1*, *Muc2*, *ZO-1* and *Cldn-1* in the presence of EHEC challenge. For each of the four HMO blends tested, an increase in *Muc1* (Figure 3A) and *Muc2* (Figure 3B) was observed, compared to EHEC O157:H7 challenge in the absence of HMOs. The expression of the intercellular tight junction protein *ZO-1* was increased by each of the four HMOs in polarized monolayers challenged with the enteric bacterial pathogen (Figure 3C). In contrast, there was no change in the expression of *Cldn-1* (Figure 3D).

To validate these transcriptional changes, the protein expression of ZO-1 and claudin-1 was then measured. As shown in Figure 3E, ZO-1 was increased by HMOs from individuals 1 to 4, but this increase was abrogated following challenge with EHEC. In addition, HMOs from individuals 2 and 3 increased claudin-1 in the absence of EHEC, but following EHEC O157:H7 challenge, claudin-1 was preserved only in cells treated with HMOs prepared from individuals 3 and 4 (Figure 3F). Taken together, these results demonstrate that while individual HMO blends share similar effects on mRNA expression of two intercellular tight junction proteins, there is variability in how HMOs prepared from different mothers preserve ZO-1 and claudin-1 protein expression in response to intestinal injury following challenge with EHEC O157:H7.

### 3.4. Individual HMO Blends Vary in Their Ability to Alter Host Inflammatory Responses

To further interrogate the effects of individual HMO blends on intestinal epithelial cells, we employed Western blotting to determine the phosphorylation of ERK1/2 and P38 MAP kinase following the challenge of intestinal cells with EHEC O157:H7. As shown in Figure 4A, cells pre-treated with each of the individual HMO preparations inhibited ERK1/2-phosphorylation, whereas the phosphorylation of P38 was inhibited by the HMOs prepared from individuals 1, 2 and 4 (Figure 4B).

EHEC O157:H7 challenge triggered the degradation of the inhibitor of kappa B alpha (IκBα), which was not prevented by any of the HMO blends (Figure 4C). To confirm these changes in NFκB signaling, we next compared downstream chemokine expression in cells pre-treated with the individual HMO blends. Challenge with EHEC increased the expression of the neutrophil-enhancing chemokine *Il-8*. Interestingly, this effect was ameliorated only by the HMOs prepared from individuals 2 and 4 (Figure 4D). In contrast, HMOs prepared from all four individuals increased the expression of the pro-inflammatory cytokine *Il-18* (Figure 4E) and the anti-inflammatory cytokines *Tgf**β* (Figure 4F) and *Il-10* (Figure 4G). HMOs prepared from milk provided by individual #1 produced the highest levels of *Il-10*. Taken together, these results demonstrate that individual HMO blends can have varying effects on selected signal transduction responses in intestinal epithelial cells.

## 4. Discussion

The concentration and compositional diversity of HMOs varies greatly in human milk between individuals. We sought to address whether such inter-individual variability has biological relevance in the intestinal tract. To begin to address this unanswered question, herein, we conducted a reductionist in vitro comparison across four HMOs prepared from randomly selected individuals and then tested for their impact on: (a) the barrier integrity of a polarized intestinal epithelial monolayer, (b) macromolecule permeability, (c) immune signaling and (d) the production of a chemokine and cytokines. In the experiments reported herein, we demonstrate that HMOs from individual mothers exert varying effects on host cell signaling responses, despite their comparable effects on maintaining the integrity of intestinal epithelial barrier functions.

HMOs are reported to exert a number of protective effects in the intestinal tract, including the stimulation of the growth of commensal bacteria [24,25], inhibition of the growth of enteric pathogens and elaborated toxins [13,26], promotion of goblet cell-derived mucin production and secretion [27], modulation of both innate and adaptive immunity [28] and maintenance of the integrity of the epithelial cell barrier [17,19]. Traditionally, the structure and specificity of responses of HMOs are associated with effects on microbial functions related to their role either as prebiotics or anti-adhesive molecules [3]. For instance, HMOs selectively enrich for bacteria, such as some lactic-acid producing bacteria and *Bifidobacteria* species that code for specific glycosidases, which permit the microorganism to metabolize the specific oligosaccharides and use the available sugars as substrates for the growth and persistence of gut colonization [24]. On the other hand, opportunist enteric pathogens are generally poor metabolizers of HMOs because they do not possess the genes necessary to produce specific glycosidases. Neutral HMOs, such as lacto-N-tetraose (LNT) and lacto-N-neotetraose (LNnT), specifically enrich for *B. breve* [10,29], whereas fucosylated HMOs enrich for other *Bifidobacteria* species [30]. Using a gnotobiotic mouse model, Charbonneau et al. previously showed that the consumption of sialic-acid containing milk oligosaccharides also specifically altered the transcriptional responses involved in growth and metabolism in *E. coli* and *B. fragilis*, suggesting a structure-specific influence of HMOs on microbial function [14]. For anti-microbial effects, 2′-fucosyllactose is uniquely efficient in blocking the binding of *Campylobacter jejuni* [31], respiratory syncytial virus [32] and rotavirus [33,34], whereas 3′-sialyllactose reduces binding of *Helicobacter pylori* to HuTu-80 gastric epithelial cells [35].

In contrast, structure–function studies comparing direct epithelial mechanisms are limited. Previous studies describe the specificity of 2′-fucosyllactose and 6′-sialyllactose related to the proliferation of HT-29 and Caco-2 intestinal epithelial cells [36,37], wherein 2′-fucosyllactose reduced levels of pro-inflammatory cytokines [38] and enhanced epithelial barrier integrity [39], and 3-fucosyllactose increased the expression of goblet cell-derived mucins and antimicrobial peptides [27]. Most recently, 2′-fucosyllactose was demonstrated to preserve intestinal epithelial integrity by protecting against cellular apoptosis [40]. Clinically, the correlation between HMOs and the intestinal epithelial barrier relates to both the diversity and presence of specific members. For instance, the low diversity of HMO composition has been reported as a risk factor for the development of necrotizing enterocolitis in extremely low birth weight infants [41]. Similarly, the presence of disialyllacto-N-tetraose in human milk also has been reported as a potential biomarker for assigning the risk of developing necrotizing enterocolitis in preterm infants, with high rates of morbidity and mortality [42]. This oligosaccharide has also been associated with the prevention of necrotizing enterocolitis, which might explain why the enteral delivery of human milk both in relevant animal models [15,19] and in human preterm babies reduces the risk of the disease [15,16]. Our results were comparable in barrier protection across the HMO composition, irrespective of diversity among the individuals.

In addition, the comparison of effects across fucosylated HMOs in our system is intriguing. Secretors (*Se+*) express the enzyme 2-fucosyltranferase (FUT2), which enables the creation of α(1-2)-fucosylated HMOs, whereas non-secretors (*Se-*) are deficient in FUT2 and express a lower abundance of α(1-2)-fucosylated HMOs [3]. Recently, a study in Bangladesh showed that approximately one third of women were identified as non-secretors [43]. In our samples, we did not identify any significant functional differences between individuals in the context of intestinal epithelial barrier function, despite variations in LNT and 2′-FL. There were comparable levels of TER and dextran flux across both groups (individuals 1 and 4 versus individuals 2 and 3). However, it is important to note that all individual samples were pasteurized to exclude microbial-mediated impacts. Recently, non-secretor status was correlated with altered milk microbiota with a decreased abundance of Lactobacillus, Enterococcus and Streptococcus species [44]. Further studies using un-pasteurized samples may be warranted to elucidate these effects.

The decoupling of the effects of HMOs on epithelial barrier function versus host inflammatory responses is intriguing. While each of the four HMO blends tested maintained TER and decreased dextran flux in response to EHEC O157:H7 challenge, there were differences in the effects on P38 MAPK signaling and *Il-8* production. In contrast, across each of the four HMOs, there were comparable changes in levels of mucins, *ZO-1* and *Cldn-1* at both the mRNA and protein expression levels. Interestingly, a comparable expression of *Tgfβ*, a cytokine known to regulate tight junction expression and promote barrier integrity during cellular injury, was also observed [45]. These observations confirm previous studies, where it has been postulated that epithelial barrier protection is a core function, which is common in various HMOs. For instance, Natividad et al. showed, using a mixture of fucosylated, sialylated and acetylated synthetic HMOs at physiologic concentrations, an increase in TER in polarized epithelial (Caco-2 and HT-29) cells, whereas exposure to any one oligosaccharide alone did not result in an increase in TER [17]. HMOs may also act to promote gut homeostasis via the induction of mucin production [19] and enhanced glycocalyx expression [39]. For instance, using reductionist model systems comparable to the present study, Kong et al. showed that fucosylation (HMOs 2′-FL and 3′FL) increases the thickness of the intestinal glycocalyx [46].

In contrast to the generalized effects of HMOs on the maintenance of epithelial barrier integrity, we observed more heterogeneous effects on host cell inflammatory responses. For instance, while all four of the HMOs tested blocked ERK1/2 signaling, none of the HMOs blocked NF-κB signaling. For downstream cytokine expression, only certain HMO preparations were able to reduce EHEC O157:H7-induced levels of *Il-8*, a chemokine previously correlated with the severity of NEC [47]. This finding fits with the concept in which immune responses to HMOs are highly specific [19,46,48]. For instance, 2′-fucosyllactose quenches host inflammatory response through the specific inhibition of Toll-like receptor 4 and the alteration of CD-14 expression [38,49] and NFκB [40]. On the other hand, 3-fucosyllactose, lacto-N-neotetraose and lactodifucotetraose reduce inflammation by blocking the activation of TNF-α [28]. Sialyated HMOs have been described to increase host susceptibility to colitis [50] and improve cognitive function in vivo [51].

In the experiments reported herein, the extent of reduction in cytokine levels varied in response to various HMO preparations and appeared to be unrelated to the activation of NFκB signaling. This finding contrasts with the report by Sodhi et al., where an inhibition of TLR-4 and NFκB activation in response to exposure to both 2′-fucosyllactose and 6′-sialyllactose was observed [52]. These differences might be related to differences in the administration of single oligosaccharide structures present in HMO, compared with the use of a pool of HMOs that are normally present in human milk. The low expression of TLR-4 in the Caco-2 cell-line is another potential explanation that could underlie the disparate findings [53].

Our observation of mechanistic variations among individual HMO blends has several important implications. First, it suggests the concept of “core” versus “additive” functions of HMOs where we observed that the barrier–protective effects of the oligosaccharides were common in most mothers, whereas immune regulation was both individual- and pathway-specific. Second, as infant formulas are increasingly being supplemented with synthetic oligosaccharides mimicking those present in human milk [54,55], a better understanding of the mechanistic variations between oligosaccharides could well prove pivotal to the design of improved preparations of infant formula. For instance, supplementation with disialyllacto-N-tetraose is reported to provide protection against the development of necrotizing enterocolitis in an animal model [15]. Therefore, in circumstances in which human milk is not available for use, the supplementation of formula for preterm infants might prove more effective if disialyllacto-N-tetraose were to be added to the preparation.

We acknowledge several limitations of our study. First, determining compositional analysis by using HPAEC–PAD is a methodology that identifies only 16 of the major HMOs that are normally found in human milk samples. Therefore, this technology does not capture the entire spectrum of oligosaccharides normally present in human milk. It is likely that less common, but still potentially bioactive, HMOs are not sufficiently accounted for in such an analysis. Second, Caco-2 cells are an immortalized, cancer-derived intestinal cell line that grows well and reproducibly in tissue culture and is physiologically analogous to absorptive enterocytes [56]. Future experiments using non-transformed cells including, for example, primary human-derived organoids and enteroids, and in vivo models are warranted to confirm the findings described herein. Third, the impact of HMOs on the growth and infectivity of the enteric bacterial pathogen EHEC O157:H7 was not tested. Further studies to characterize whether bacterial growth and the expression of virulence genes changes following exposure to various HMO preparations are warranted. Lastly, the inherent differences identified were derived from four individual donors, and do not capture other major differences in HMO blends, including Secretor and Lewis status. Future studies comparing the direct cellular effects of greater numbers of donors are warranted to substantiate our findings.

In conclusion, using an EHEC O157:H7-induced reductionist model of intestinal injury, we have shown that HMOs prepared from human milk protect against pathogen-induced changes in intestinal epithelial barrier integrity and macromolecular permeability. In contrast, we describe differences in host cell signal transduction responses and the expression of innate and adaptive immune responses following exposure to compositionally distinct HMO preparations of human milk. Taken together, these results provide evidence of inter-individual variations in direct HMO-mediated effects on the intestinal epithelium, which may be an important factor explaining why exclusively breastfed preterm infants develop NEC, despite the overall preventative effects of breast milk.

## Figures and Tables

**Figure 1 nutrients-14-01014-f001:**
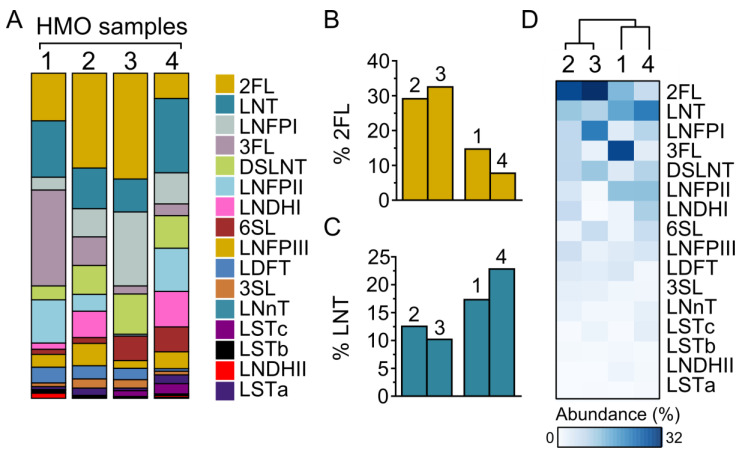
Composition of HMOs prepared from four individuals. (**A**) Percentage composition of HMOs extracted from four mothers (#1, 2, 3 and 4) determined by using HPAEC–PAD. (**B**–**C**) HMO composition of mothers #2 and #3 versus mothers #1 and #4 differ in the abundance of both LNT and 2′-FL. (**D**) HMO composition across four maternal donors clustered by STAMP using the unweighted pair group method with arithmetic mean. 2FL, 2′-fucosyllactose; 3FL, 3′-fucosyllactose; LDFT, lacto-difucotetraose; LNT, lacto-N-tetraose; LNnT, lacto-N-neotetraose; LNFPI, lacto-N-fucopentaose I; LNFPII, lacto-N-fucopentaose II; LNFPIII, lacto-N-fucopentaose III; LNDHI, lacto-N-difucohexaose I; LNDHII, lacto-N-difucohexaose II; 3SL, 3′-sialyllactose; 6SL, 6′-sialyllactose; LSTa, sialyl-lacto-N-tetraose a; LSTb, sialyl-lacto-N-tetraose b; LSTc, sialyl-lacto-N-neotetraose c; DSLNT, disialyl-lacto-N-tetraose.

**Figure 2 nutrients-14-01014-f002:**
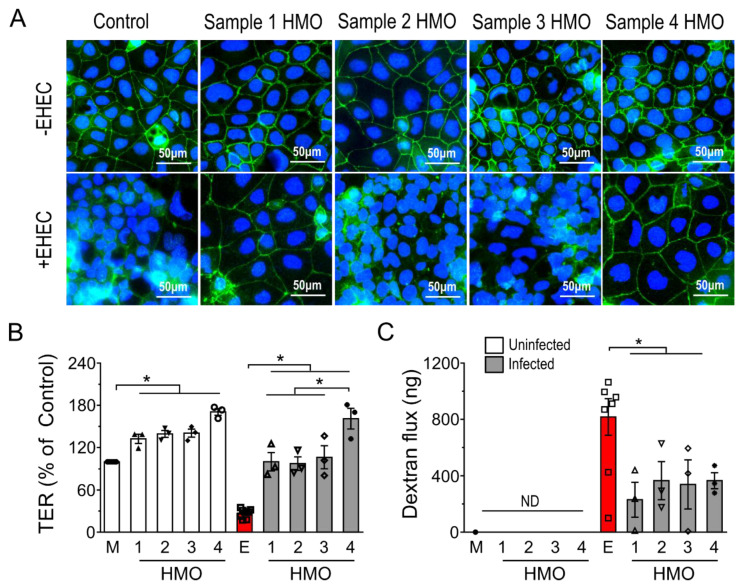
Functional effects of HMOs from four individuals on Caco-2Bbe1 epithelial barrier integrity. (**A**) En face images of immunofluorescence microscopy showing the intercellular tight junction protein ZO-1 in Caco-2Bbe1 cells either in the absence of HMOs (control) or pre-treated (20 mg/mL; 24 h) with four individual HMO preparations alone (-EHEC) or followed by EHEC O157:H7 challenge (4 h, MOI 100:1). Green denotes ZO-1 and blue DAPI nuclear staining. Images are representative of 3 separate experiments. (**B**) Transepithelial electrical resistance (TER), expressed as percentage of PBS treatment used as control, was measured in polarized Caco-2Bbe1 epithelial cells grown on semi-permeable polyester Transwell filters incubated with HMOs prepared from each of four individuals, followed by 4 h EHEC O157:H7 challenge (MOI: 100 to 1; n = 4–5 per group). (**C**) Caco-2Bbe1 cells in Transwells were then washed with PBS and exposed to 10 kDa dextran in the apical compartment (5 h) and flux of the fluoresceine-labelled macromolecule then measured in the basolateral compartment (n = 4–5/group). Bars represent means, ±SEM; * denotes *p* < 0.05 (ANOVA with Bonferroni post hoc analysis). M denotes medium; E denotes EHEC; numbers 1 through 4 denote HMO composition preparations from individuals 1 to 4.

**Figure 3 nutrients-14-01014-f003:**
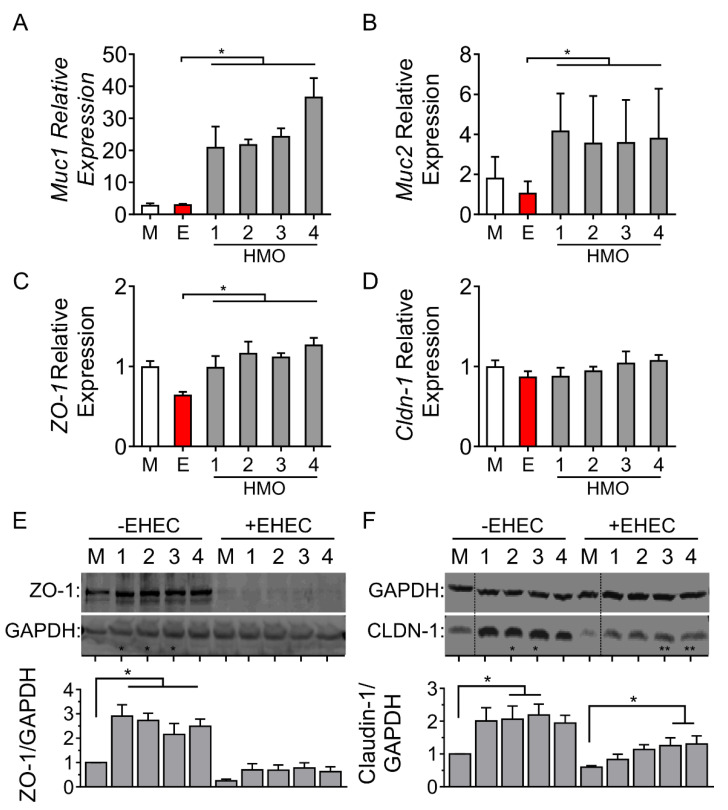
Effects of individual HMO blends on the expression of innate defense and intercellular tight junction proteins in Caco-2Bbe1 cells. Epithelial cells grown in 24-well plates were treated with each of the HMOs prepared from four individuals followed by EHEC O157:H7 challenge (4 h, MOI 100:1). mRNA was extracted to measure, using RT-PCR, levels of: (**A**) *Muc1*, (**B**) *Muc2*, (**C**) *Zo-1* and (**D**) *Cldn-1* (n = 3–4 per group). Western blotting shows protein levels of (**E**) ZO-1 and (**F**) Claudin-1 after 4 h of EHEC challenge with corresponding densitometry of immunoblots shown below each representative immunoblot (n = 4 per group). Densitometry ratios were measured using GAPDH as a loading control. The dotted line indicates spliced sites from a single original immunoblot. Values are expressed as means, ±SEM; * denotes *p* < 0.05 (ANOVA with Bonferroni post hoc analysis). M denotes medium; E denotes EHEC; numbers 1 through 4 denote HMO composition preparations from individuals 1 to 4.

**Figure 4 nutrients-14-01014-f004:**
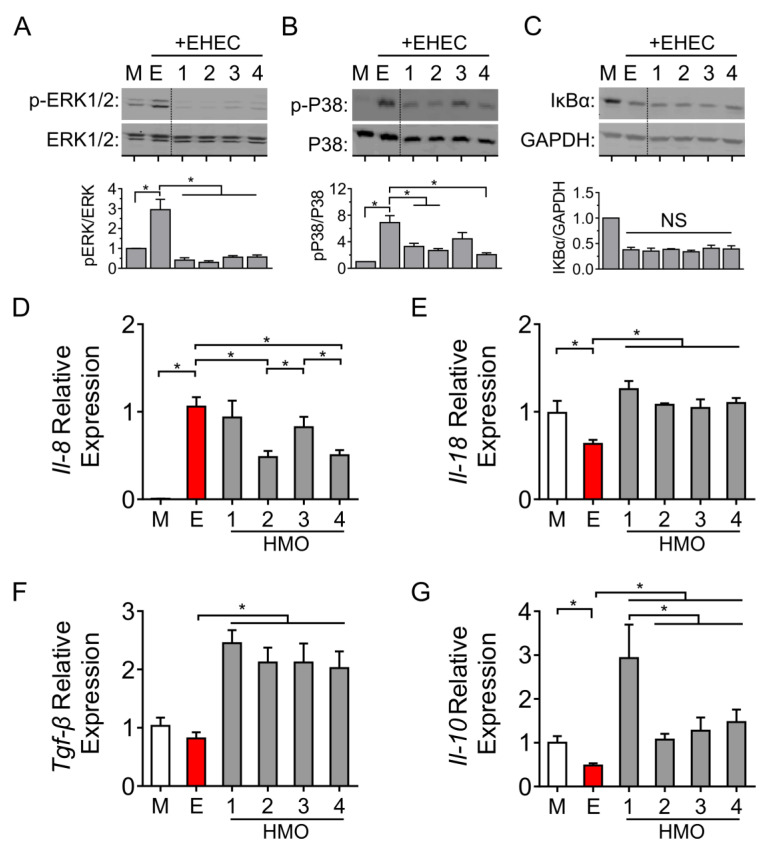
**Effects of individual HMO blends on immune signaling and chemokine/cytokine expression in Caco-2Bbe1 cells.** Epithelial cells grown in 24-well plates were pre-treated with each of the four HMO preparations followed by EHEC O157:H7 challenge (3 h, MOI 100:1) and then assessed for (**A**) phosphorylation of ERK1/2 (n = 4–5 per group), (**B**) phosphorylation of P38 MAP kinase (n = 4–5 per group) and (**C**) NF-κB activation (n = 4 per group). Densitometry ratios were measured using corresponding non-phosphorylated MAPK or GAPDH as loading controls. Dotted lines indicate spliced sites from a single, original Western blot. At 4 h post-EHEC challenge, mRNA was extracted to measure levels of (**D**) *Il8*, (**E**) *Il18*, (**F**) *Tgfβ* and (**G**) *Il10* (n = 3–4 per group). Data are expressed as means, ±SEM; * denotes *p* < 0.05 (ANOVA with Bonferroni post hoc analysis). M denotes medium; E denotes EHEC; numbers 1 through 4 denote HMO composition preparations from individuals 1 to 4.

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
