# Peer review of "Variations in the Composition of Human Milk Oligosaccharides Correlates with Effects on Both the Intestinal Epithelial Barrier and Host Inflammation: A Pilot Study"

_nutrients, 2022, doi:10.3390/nu14051014_

Round 1
Reviewer 1 Report
The study presented in this manuscript is interesting and relvant for the scientific community.
But i can see a limitation, thats why i would consider the study only a preliminary study and the results to be confirmed by futher research: donated surplus human milk samples from four individual unidentified donors - only 4 individual donors?
Please reconsider maybe as a short communication, and in discussions elaborate some further perspectives.
Reviewer 2 Report
This detailed paper showing that blends of extracted HMOs from individual mothers support nursing infant health and development directly through effects on intestinal epithelial cells and on host inflammation, is important and novel. My review suggests several small changes could improve the paper:
- On page 2 line79, the details on centrifugation should be given as relative centrifugal force, "x g", rather than in rpm, to allow for reproduction of the protocol.
- It should be clarified earlier in the paper than in the discussion of limitations, that only the 16 most prominent HMOs were measured in the 4 human milk samples, and that the results may be due to other components. Also, throughout the discussion, the blends of extracted oligosaccharides are referred to as "the four HMOs" which may give a distorting impression superficially, rather than, more properly, "HMO blends".
- On page 4 beginning of line 140, it would be more grammatically correct to insert "assessment after "Monolayer permeability ..."
- On page 5 line 186, it is stated, the samples from each mother were collected "over time"- a clearer statement of how long a period was involved, would be better.
- For Figures 3 and 4, the symbols M and E are used, but one cannot read Figure 4 independently, as the definition is only mentioned in the caption for Figure 3. If they will be printed together this is not an issue.
- On page 11 line 399, the second word, "in", is excessive and would best be removed. In line 403, page 11, after the first word "administration", the word "of" should be inserted.
Round 2
Reviewer 1 Report
The reviewer would like to thank the authors for the revised submission. I further support the publication of the manuscript.